# Quantum many-body simulations with PauliStrings.jl

Nicolas Loizeau,[1] J. Clayton Peacock,[2] and Dries Sels[2,3]

[1] *Niels Bohr Institute, University of Copenhagen, Copenhagen, Denmark*
[2] *Department of Physics, New York University, New York, NY, USA*
[3] *Center for Computational Quantum Physics, Flatiron Institute, New York, NY, USA*
(Dated: January 27, 2025)

We present the Julia package **PauliStrings.jl** for quantum many-body simulations, which performs fast operations on the Pauli group by encoding Pauli strings in binary. All of the Pauli string algebra is encoded into low-level logic operations on integers, and is made efficient by various truncation methods which allow for systematic extrapolation of the results. We illustrate the effectiveness of our package by (i) performing Heisenberg time evolution through direct numerical integration and (ii) by constructing a Liouvillian Krylov space. We benchmark the results against tensor network methods, and we find our package performs favorably. In addition, we show that this representation allows for easy encoding of any geometry. We present results for chaotic and integrable spin systems in 1D as well as some examples in 2D. Currently, the main limitations are the inefficiency of representing non-trivial pure states (or other low-rank operators), as well as the need to introduce dissipation to probe long-time dynamics.

## CONTENTS

# I.  INTRODUCTION

It is generally difficult to represent quantum objects on a classical computer because they live in a Hilbert space that is exponential in system size. Hamiltonians of discrete systems are trivially represented as matrices that take exponential space, and pure states are vectors on which these matrices can act. It is this basic fact that limits the classical simulability of quantum systems, one way or another. However, most systems of interest have a lot more structure, and this structure can often be exploited to devise more meaningful representations of the problem. Most quantum systems of interest are local, i.e. subsystems with few degrees of freedom typically interact through few-body interactions on a geometrically local graph. This locality can be taken advantage of; for example, it has been proven that gapped Hamiltonians have low energy eigenstates which obey the 'area law', that is, their entanglement entropy in a given region of space scales as the area of the boundary of that region rather than as the volume, which would be the case generically [1–3]. Thus, the manifold of Hilbert space in which these states live is very small with respect to the total space, and in addition under time evolution the amount of Hilbert space explored is exponentially tiny [4].

We can take advantage of this by choosing ansatzes to represent our states which also obey the area law. Tensor networks such as Matrix Product States (MPS) [5–7] are one such set of ansatzes which have been shown to be extremely effective at representing quantum states of local Hamiltonians [8]. Operators can also be represented as tensor networks, such as in the form of Matrix Product Operators (MPO) [9, 10].

In this work we will explore a different representation for efficiently modeling such systems; for example when we write the Hamiltonian of an Ising chain

$$H = \sum_i Z_i Z_{i+1} + \sum_i X_i \tag{1}$$

we take advantage of the locality of $H$ to decompose it into local terms : $H = \sum_i \tau_i$ where the $\tau_i$'s are Pauli strings. Each Pauli string is the tensor product of a Pauli matrix $\{\mathbb{1}, X, Y, Z\}$ at each site. Since these strings form a complete basis, all operators in the Hilbert space can simply be encoded as linear combinations of Pauli strings.

In this work, we will show that this encoding can be advantageous for numerical simulation of quantum dynamics. The advantage arises from two key features:

1. The Pauli string algebra is encoded in low-level logic operations on integers, making it very efficient to numerically store and multiply strings together.

2. Operators can be systematically truncated to some precision by discarding strings with negligibly small weight. This allows one to keep the number of strings manageable at the cost of some incurred error.

The second point is particularly powerful when combined with noise. It has recently been shown that noisy quantum circuits can be simulated in polynomial time [11, 12]. The proof is based on the idea that long Pauli strings decay exponentially in time with an exponent that is proportional to their length. Because there are only polynomially many short strings, truncation of the long strings makes simulations more tractable. This can be seen as a truncation of the very non-local correlations. Note that here 'non-locality' refers to many body interactions, not to 'geometric non-locality'. Thus, the method does not truncate geometrically non-local correlations if they are encoded in few body terms. The Pauli string representation is therefore the natural way to take

advantage of noise to classically simulate quantum many body systems. To make this practical, we present a user-friendly Julia package that easily encodes local Hamiltonians and operators, as well as implements important techniques in the study of quantum many-body systems: **PauliStrings.jl**.

The paper is organized as follows. We will first outline the numerical methods. Then, we will show results obtained with **PauliStrings.jl** for Heisenberg time evolution and Krylov subspace expansion (the recursion method) of operators in 1D, 2D, integrable, and chaotic systems. To benchmark our method, we also present results obtained with tensor network techniques. Since we are interested in high-temperature dynamics, we find that the Pauli string algebra outperforms tensor networks in a number of cases. Finally, we discuss ways in which the Pauli string method can still be improved.

## II. METHODS

### Pauli strings

To encode the algebra of Pauli strings in logic operations on binary strings, we utilize the method laid out in Ref. [13]. Here, we give an overview of this encoding and show how we use it to efficiently manipulate quantum operators. First define the following real matrices

$$\tau_{00} = \mathbb{1}_2 \tag{2}$$
$$\tau_{01} = X \tag{3}$$
$$\tau_{10} = Z \tag{4}$$
$$\tau_{11} = iY \tag{5}$$

where $X, Y, Z$ are the Pauli matrices. Up to a phase $\alpha$, we can multiply two $\tau$ matrices by performing two XOR operations on their indices:

$$\tau_{v_1 w_1} \tau_{v_2 w_2} = \alpha \tau_{(v_1 \oplus v_2)(w_1 \oplus w_2)}. \tag{6}$$

We can use this property to efficiently multiply Pauli strings. Encode a Pauli string $\tau_a$ in a tuple of binary integers $a = (v, w)$ such that $\tau_a = \tau_{v_{(1)} w_{(1)}} \otimes \tau_{v_{(2)} w_{(2)}} \otimes \tau_{v_{(3)} w_{(3)}} ...$ where $v_{(i)}$ is the $i^{th}$ bit of integer $v$. Then the following relation holds:

$$\tau_{a_1} \tau_{a_2} = \alpha \tau_{a_{12}}, \tag{7}$$

where $a_1 = (v_1, w_1)$, $a_2 = (v_2, w_2)$, $a_{12} = (v_{12}, w_{12})$ and

$$v_{12} = v_1 \oplus v_2 \tag{8}$$
$$w_{12} = w_1 \oplus w_2 \tag{9}$$
$$\alpha = (-1)^{\text{pop}(v_1 \wedge w_2)}. \tag{10}$$

Here $\oplus$ denotes bitwise XOR, $\wedge$ denotes bitwise AND and $\text{pop}(n)$ counts the number of set bits of $n$ (Hamming weight or popcount). A similar relation holds for the commutator of two Pauli strings:

$$[\tau_{a_1}, \tau_{a_2}] = \alpha \tau_{a_{12}} \tag{11}$$

where (8) and (9) still hold but (10) is replaced by

$$\alpha = (-1)^{\text{pop}(v_1 \wedge w_2)} - (-1)^{\text{pop}(w_1 \wedge v_2)}. \tag{12}$$

We can now use this encoding to represent an operator as a list of tuples $\{a_i = (v_i, w_i)\}$, together with a list of complex coefficients $\{h_i\}$. The full operator is simply

$$H = \sum_i h_i \tau_{a_i}. \tag{13}$$

Computing the product or the commutator of two operators in this representation is equivalent to multiplying the coefficients and the Pauli strings $\tau_{a_i}$ two by two, as shown in algorithm 1. Algorithm 1 can also be adapted with eq. (12) in order to compute the commutator.

---

**Algorithm 1** Product of two operators $A$ and $B$ in the binary Pauli string representation. Operators are encoded as dictionaries of complex numbers indexed by tuple of integers $(v, w)$ that represent a Pauli string. In this algorithm, $A_{v,w}$ denotes the coefficient in front of the string encoded by $v, w$ in operator $A$.

---

   $C \leftarrow$ empty dictionary
   **for** $(v_A, w_A)$ in $A$.keys **do**
      **for** $(v_B, w_B)$ in $B$.keys **do**
         $v \leftarrow v_A \oplus v_B$
         $w \leftarrow w_A \oplus w_B$
         $h \leftarrow A_{v_A,w_A} \cdot B_{v_B,w_B} \cdot (-1)^{\mathrm{pop}(v_A \wedge w_B)}$
         **if** $(u, v)$ is in $C$ **then**
            $C_{v,w} \leftarrow C_{v,w} + h$
         **else**
            $C_{v,w} \leftarrow h$
         **end if**
      **end for**
   **end for**
   **return** $C$

---

In the special case that the operators have translation symmetry, there is an even more efficient way to encode them. Consider for example the 1D Ising Hamiltonian with periodic boundary conditions $H = -J(\sum_i Z_i Z_{i+1} + g \sum_i X_i)$. In this case, there is no need to store all the Pauli strings. $H$ is fully specified by the two strings $-J Z_1 Z_2$ and $-J g X_1$ and the fact that it has translation symmetry. In general, a 1D translation symmetric operator can be written as $\sum_i T_i(H_0)$ where $T_i$ is the $i$-site's translation operator and $H_0$ is the local operator that generates $H$. $H_0$ can be chosen so that it's only composed of Pauli strings that start on the first site. Algorithm 2 shows how to take the product of two 1D translation symmetric operators. The main difference from Algorithm 1 is that we need to translate each string back so that it starts on the first site. In **PauliStrings.jl**, this is implemented in the `OperatorTS1D` structure. Note that Algorithm 2 can easily be adapted to higher dimensions by iterating over the necessary shifts corresponding to each dimension.

In both cases, (translation symmetric or not), the numerical power of this representation lies in the possibility to efficiently truncate an operator by only keeping Pauli strings with the largest weights. When running iterative algorithms like Lanczos, or discrete time evolution, we truncate the operator at each step by keeping a maximum number of strings or by discarding long strings.

---

**Algorithm 2** Product of two translation symmetric operators $A$ and $B$ supported on $N$ spins. $T_k$ is the translation operator by $k$ sites and $\mathrm{Shiftleft}(v, w)$ translates the string $(v, w)$ such that it starts on the first site.

---

$C \leftarrow$ empty operator
**for** $(v_A, w_A)$ in $A$.keys **do**
    **for** $(v_B, w_B)$ in $B$.keys **do**
        **for** $0 \leq k < N$ **do**
            $(v'_B, w'_B) \leftarrow T_k(v_B, w_B)$
            $(v, w) \leftarrow (v_A \oplus v'_B, w_A \oplus w'_B)$
            $(v, w) \leftarrow \mathrm{Shiftleft}(v, w)$
            $h \leftarrow A_{v_A, w_A} \cdot B_{v_B, w_B} \cdot (-1)^{\mathrm{pop}(v_A \wedge w_B)}$
            **if** $(v, w)$ is in $C$ **then**
                $C_{v,w} \leftarrow C_{v,w} + h$
            **else**
                $C_{v,w} \leftarrow h$
            **end if**
        **end for**
    **end for**
**end for**
**return** $C$

---

### Tensor networks

We will use tensor networks to benchmark **PauliStrings.jl**, because as mentioned previously, they also provide a powerful tool to work around the exponential size of the Hilbert space (for a review of tensor networks we refer readers to Refs. [14, 15]). By constructing a tensor network, one can compress a state or operator living in the exponentially large Hilbert space into a polynomial number of chain of tensors, which when contracted, recover the full state/operator. For this representation to be efficient, the number of tensors should be polynomial in the system size.

Matrix Product States (MPS) are 1D tensor networks which can be used to represent the quantum state of $N$ spins by constructing $N$, 3-dimensional tensors; two of the indices run over an internal 'bond-dimension' (BD) and the third index represents the spin degrees of freedom. Matrix Product Operators (MPO) are similarly used to represent quantum operators and consist of $N$ tensors, each with 4-dimensions (two bond-dimensions and two spin degrees of freedom). When doing calculations a maximum bond-dimension of tensors can be set, inducing a controlled error in the representation of the state/operator, but at the benefit of reducing the number of parameters to be just polynomial in the system size.

Using these truncations effectively, tensor networks are highly efficient at performing matrix operations and time evolution with a local and highly sparse Hamiltonian $H$. In this work, we will use the **ITensors.jl** julia package [16] to run all tensor network simulations, and we refer readers to its documentation for a precise definition of the 'cutoff' parameter we use in our simulations. Heisenberg time evolution is performed with the operator form of the Time Evolving Block Decimation (TEBD) algorithm [17, 18] as is implemented in **ITensors.jl**, and we perform the tensor network recursion method simulations described later by representing all operators in the algorithm as MPOs.

## III.   HEISENBERG TIME EVOLUTION

Computing time evolution with the Pauli strings is done in the Heisenberg picture. Indeed, a pure state is a low-rank density matrix, and low-rank matrices cannot be efficiently encoded as the sum of Pauli strings [19]. It is therefore more efficient to evolve a local operator than a pure state in the Pauli strings representation. Here this is done by integrating Von Neuman's equation

$$i\frac{dO}{dt} = -[H, O] \tag{14}$$

using the Runge-Kutta method. To keep the number of strings manageable, we introduce noise and truncate the operator $O$ at each time step by keeping the strings with the largest weight. The noise is modeled by a depolarizing channel that causes long Pauli strings to decay. In the Heisenberg picture, observables evolve under the adjoint channel. Because a depolarizing channel is self-adjoint, we can apply it directly to $O$. The transmission rate associated with a Pauli string of length $w$ is $e^{-\epsilon w}$ where $\epsilon$ is the noise amplitude. Similar approaches have been recently used in Refs. [11, 20–25]. The strategy here is to choose the smallest noise value that makes the simulations tractable, while not destroying the phenomena of interest.

### Results: Next-nearest neighbor XXZ chain

As an example, we discuss the diffusion of a local operator in a XXZ next-nearest-neighbor spin chain

$$H = \sum_i \left( X_i X_{i+1} + Y_i Y_{i+1} + \Delta Z_i Z_{i+1} \right) \tag{15}$$
$$+ \gamma \sum_i \left( X_i X_{i+2} + Y_i Y_{i+2} + \Delta Z_i Z_{i+2} \right)$$

with $\gamma = \frac{1}{2}$ and $\Delta = 2$. In **PauliStrings.jl** we can build this Hamiltonian as follows:

```
function XXZnnn(N::Int)
    Δ = 2
    γ = 1/2
    H = ps.Operator(N)
    for j in 1:N
        H += "X",j, "X",j%N+1
        H += "Y",j, "Y",j%N+1
        H += Δ, "Z",j, "Z",j%N+1
        H += γ, "X",j, "X",(j+1)%N+1
        H += γ, "Y",j, "Y",(j+1)%N+1
        H += γ*Δ, "Z",j, "Z",(j+1)%N+1
    end
    return H
end
```

where the modulo ensure periodic boundary conditions.

It is known to be difficult to numerically recover the hydrodynamic diffusive behavior of strongly coupled spin chains, with some of the best current methods being the truncated Wigner approxima-

tions [26, 27] and TEBD [28]. Diffusion can be observed as a $\sim \frac{1}{\sqrt{t}}$ decay of the infinite-temperature autocorrelation function:

$$S(t) = \frac{1}{2^N} \text{Tr}[Z_1(t)Z_1(0)]. \tag{16}$$

We recover this behavior with our truncated Pauli string simulations as shown in Fig. 1 for relatively large system size $N = 30$, where the grey lines indicate $\sim \frac{1}{t^{-1/2}}$ scaling and $M$ denotes the maximum allowed number of Pauli strings. All results were computed within 1 day maximum runtime.

Our method allows us not only to compute $S(t)$ but also more general two-time correlation functions between $Z_1$ and other Pauli strings. To illustrate this, in Fig. 2 we plot the two point correlator:

$$S_{i-j}(t) = \frac{1}{2^N} \text{Tr}[Z_i(t)Z_j(0)]. \tag{17}$$

In the diffusive regime, we'd expect these correlations to decay like a Gaussian with a width that grows like $\sigma^2 \sim t$. As the depolarizing noise $\epsilon$ is increased, the results increasingly converge in increasing $M$ to the expected diffusive decay. However, for the highest values of $\epsilon$, the effects of the breaking of energy and particle-number conservation start to manifest itself. The system then locally relaxes to equilibrium, resulting in a crossover from diffusion to exponential decay. By carefully choosing a moderate value of $\epsilon$ and large $M$, one can see good convergence to the expected diffusive decay up to large $t$ (see in particular the plot for $\epsilon = 0.01$). In addition, by scaling out the particle loss $n(t) = \sum_j \text{Tr}[Z_0(t)Z_j(0)] = e^{-\epsilon t}$ to correct for the purely dissipative effect that comes from the depolarizing channel, one observes a broad regime of $\epsilon$ and $t$ over which the results converge, as shown in Fig. 3.

The following is a code example of noisy time evolution implementation in **PauliStrings.jl** used to generate Fig. 1:

```julia
# heisenberg evolution of the operator O using rk4
# return tr(O(0)*O(t))/tr(O(t)^2)
# M is the number of strings to keep at each step
# noise is the amplitude of depolarizing noise
function evolve(H, O, M, times, noise)
    S = []
    O0 = deepcopy(O)
    dt = times[2]-times[1]
    for t in times
        push!(S, ps.trace(O*ps.dagger(O0))/ps.trace(O0*O0))
        #preform one step of rk4, keep only M strings, do not discard O0
        O = ps.rk4(H, O, dt; heisenberg=true, M=M,  keep=O0)
        #add depolarizing noise
        O = ps.add_noise(O, noise*dt)
        # keep the M strings with the largest weight. Do not discard O0
        O = ps.trim(O, M; keep=O0)
    end
    return real.(S)
end
```

Achieving time evolution at this system size with dense or sparse matrices would require a large amount of distributed memory, making it very computationally expensive to run. On the other hand, using TEBD as shown in Fig. 4 we are able to get converged results with 9 Gb of memory but only up to $t \sim 1$ in 4 days runtime. The **PauliStrings.jl** result with $M = 2^{18}$ and $\epsilon = 0.01$ is shown for comparison and displays the expected decay to an order of magnitude longer time ($t \sim 10$), though it only required 5 Gb of memory and 1 day of runtime. The TEBD results shown use a truncation cutoff of $10^{-10}$; we also performed the same simulation for both larger and smaller cutoffs, but found $10^{-10}$ to be more than sufficiently converged while larger cutoffs did not improve accessible simulation times without significant loss of accuracy. Thus **PauliStrings.jl** performs significantly better than TEBD for Heisenberg time evolution of the next-nearest neighbor XXZ chain.

## IV.   KRYLOV SUBSPACE EXPANSION

Much recent work has shown that Krylov subspace expansions of the Liouvillian, through the recursion method, provide valuable insights in quantum many-body dynamics beyond simply solving the equations of motion (recently reviewed in Ref. [29]). The recursion method has been used for decades to study quantum many-body systems, and is explained in detail in Ref. [30]. The method consists of utilizing the Lanczos algorithm to tridiagonalize the Liouvillian $\mathcal{L} = [H, \cdot]$, which is the superoperator which generates time evolution of operators in the Heisenberg picture as $\frac{dO}{dt} = i[H, O]$.

Lets introduce Lanczos' algorithm. The idea is to construct an orthonormal basis of operators generated by recursively applying $\mathcal{L}$ to $O$. First, one defines an inner product, which here we choose to be the Frobenius inner product (up to normalization): $AB = \frac{1}{2^N}\text{Tr}[A^\dagger B]$ and norm $\|O\| = \frac{1}{2^N}\text{Tr}[O^2]$. Then, the first iteration is given by:

$$O_1 = \mathcal{L}O_0/b_1 = [H, O_0]/b_1,$$
$$b_1 = \|\mathcal{L}O_0\|. \tag{18}$$

For $n > 2$ the algorithm proceeds as follows, up to a maximal dimension $n = D^2 - D + 1$ where $D$ is the Hilbert space dimension [31]:

$$O'_n = \mathcal{L}O_{n-1} - b_{n-1}O_{n-2},$$
$$O_n = \frac{O'_n}{b_n},$$
$$b_n = \|O'_n\|. \tag{19}$$

In the end, one has generated an orthonormal 'Krylov-basis' $\{\mathcal{L}O, \mathcal{L}^2O, ...\mathcal{L}^nO\}$ and 'Lanczos coefficients' $b_n$ which are also uniquely related to the moments of the Hamiltonian [30].

The recursion method was first used in the 1980s to approximate time evolution [32, 33], and has also been used more recently to calculate conductivities [34–39]. However in this work we will focus on even more recent developments of the recursion method as a probe of quantum chaos [38, 40–45]. The logic is as follows; evolving in time under the Liouvillian, an initially local operator becomes increasingly nonlocal and complex, requiring an increasing number of basis vectors from the Hilbert space to represent it. The Lanczos coefficients generated by the recursion method are a measure of this complexity. These coefficients are expected to grow as fast as possible in chaotic systems, which has been strictly bounded to be linear, with a logarithmic correction in one-dimension [38]. However, in integrable systems, due to the presence of conserved quantities, the dynamics is restricted and

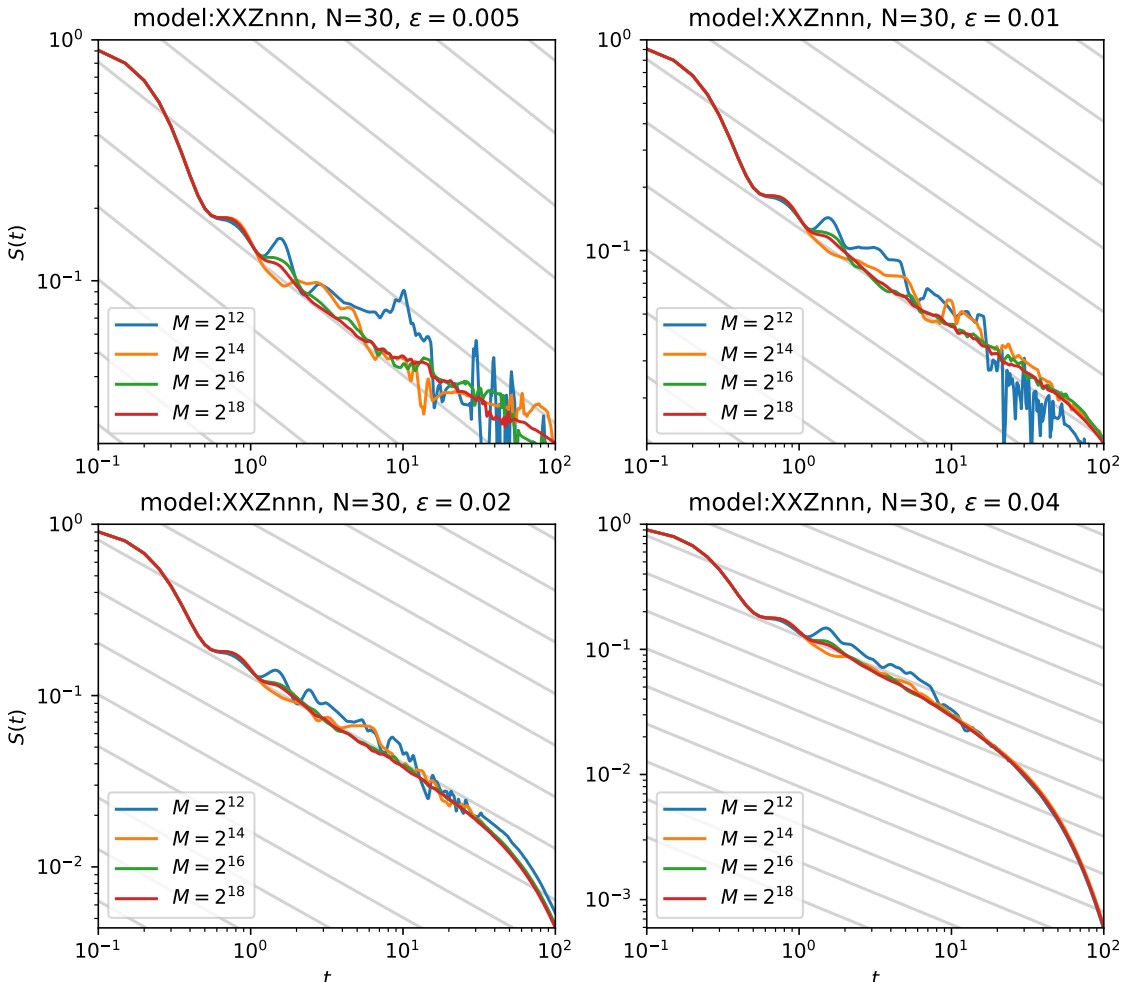

FIG. 1. Diffusive decay of correlator (16) computed by time evolving the XXZ next-nearest-neighbor spin chain (15) using the Pauli strings method. The grey lines show the $\sim \frac{1}{\sqrt{t}}$ decay. $M$ denotes the maximum number of strings in the time evolution. Only $M$ strings with the highest weight are kept at each time step of the RK4 integration.

the Lanczos coefficients are generally expected to grow sublinearly (commonly as $\sim \sqrt{n}$), and don't grow at all for systems which can be mapped to free fermions [38].

Thus the rate of growth of the Lanczos coefficients can be used as a generic probe of quantum chaos. In addition, using these ideas it has recently been shown that the knowledge of a few Lanczos coefficients can be sufficient to estimate long time dynamics [46] and to probe for hydrodynamics [28, 39].

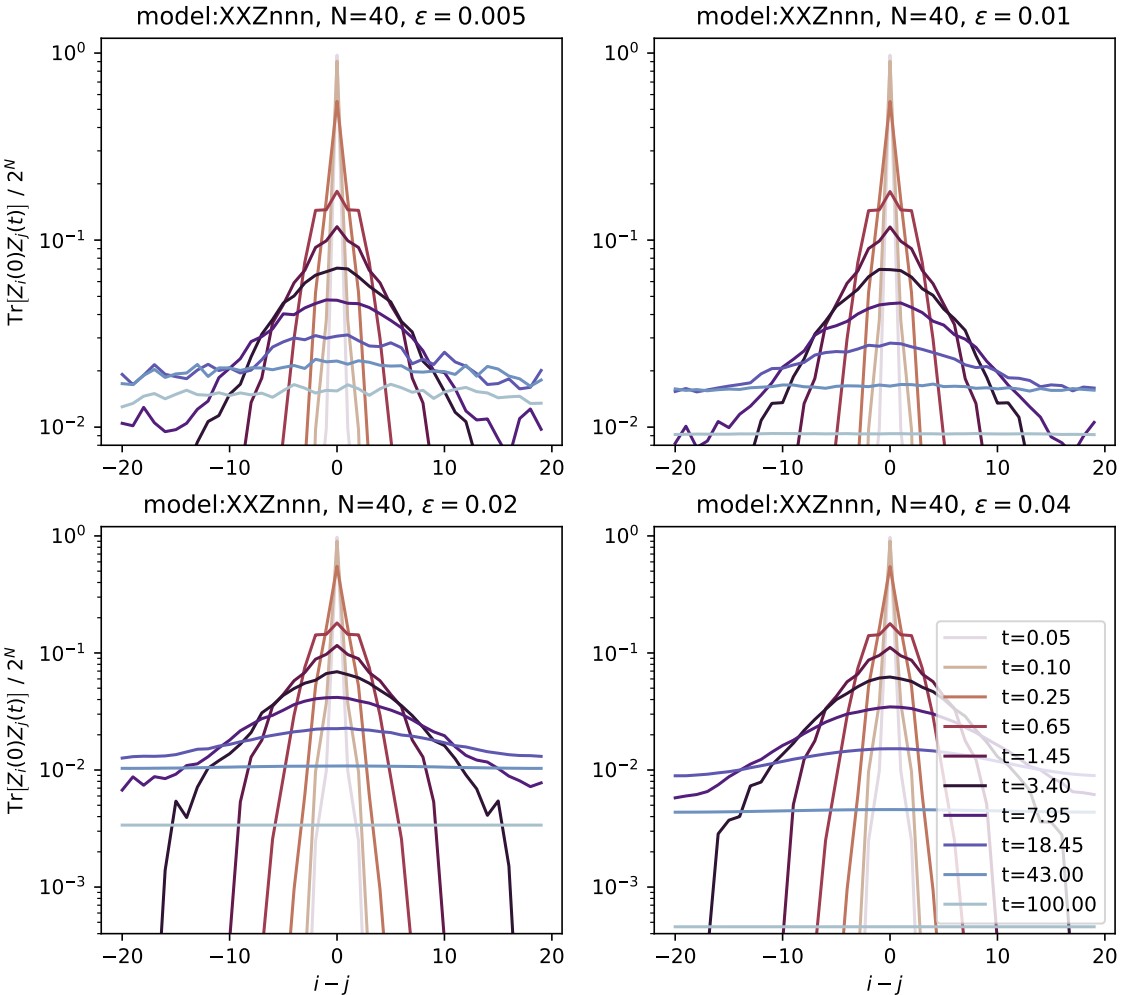

FIG. 2. Diffusive decay of the two point correlator (17). The two point correlator takes the form of a Gaussian function in $i-j$ spreading with time. This is characteristic of diffusion. Note that in our Pauli strings representation, extracting this quantity is not more computationally costly than extracting the correlator (16) from Fig. 1. The presence of noise makes the correlator decay faster than diffusive at late times, as also seen on Fig. 1.

**Results: Lanczos coefficients**

In this section we show Lanczos coefficients for different systems calculated with both the truncated Pauli string method and MPOs. In both cases we use a maximum memory of 40 Gb, however, importantly, in the tensor network simulations 4 CPUs are used while in **PauliStrings.jl** there is currently no parallelization implemented and only 1 CPU is used. Fig. 5 compares Pauli strings without exploiting translation symmetry to MPO while Fig. 7 shows Pauli strings results using

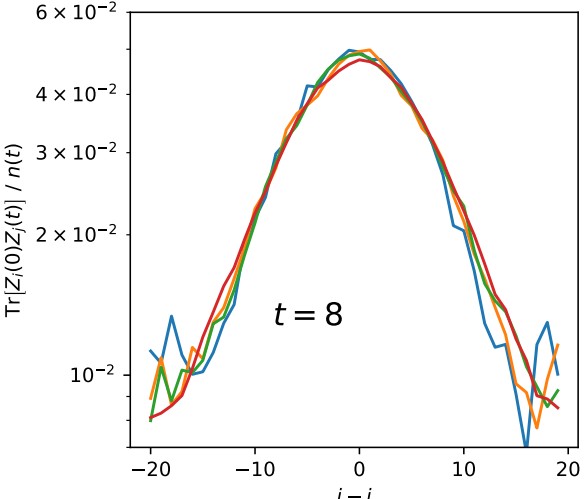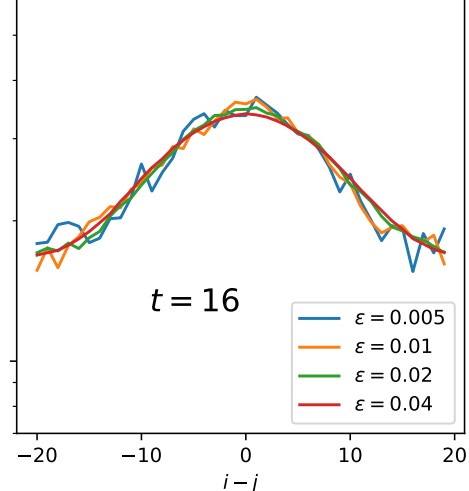

FIG. 3. Two point correlator (17) (same as in Fig. 2) for some particular times and different values of dissipation $\epsilon$. On this figure, the correlator (17) is normalized by $n(t) = \sum_j \text{Tr}[Z_0(t)Z_j(0)] = e^{-\epsilon t}$ in order to correct for the purely dissipative effect that comes from the depolarizing channel only. At short time (e.g. $t = 8$), the correlator is Gaussian while at longer times (e.g. $t = 16$), the edges of the Gaussian flatten due to finite size effects.

translation symmetry.

Integrable Models

We first consider two interacting integrable models. The universal behaviors of this class are not fully understood, however it is known that the Lanczos coefficients have square root growth $b_n \sim \sqrt{n}$ in many standard models such as those studied here [30, 38, 47].

We first consider the **XX model** with Hamiltonian:

$$H = \sum_i \left( X_i X_{i+1} + Y_i Y_{i+1} \right). \tag{20}$$

To construct this Hamiltonian in **PauliStrings.jl** with open boundary conditions one writes the following julia code:

```julia
function XX(N)
    H = ps.Operator(N)
    for j in 1:(N - 1)
        H += "X",j,"X",j+1
        H += "Y",j,"Y",j+1
    end
    return H
end
```

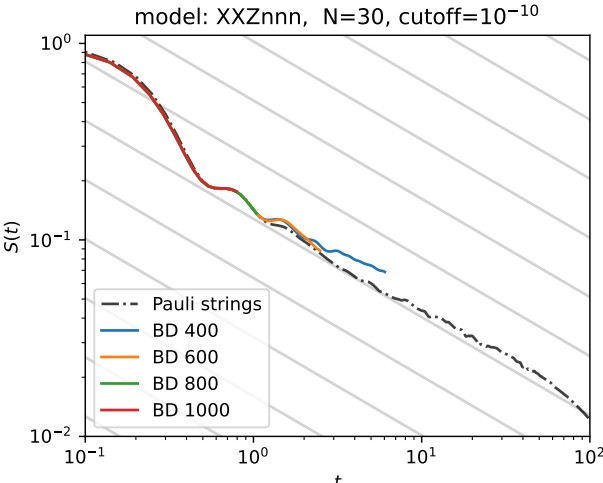

FIG. 4. Diffusive decay of the two point correlator (16) computed by time evolving the XXZ next-nearest-neighbor spin chain (15) using TEBD. The grey lines show the $\sim \frac{1}{\sqrt{t}}$ decay, BD denotes the maximum bond dimension allowed in the time evolution, and results are shown for a truncation cutoff of $10^{-10}$. The Pauli strings result with $M = 2^{18}$ and $\epsilon = 0.01$ is shown for comparison.

We then build the Krylov space from an initial operator $O_0 = \sum_{i=1}^{N} X_i$. The results for this model are shown in Fig. 5 (A) for $N = 40$ up to $n = 40$ Lanczos iterations. With **PauliStrings.jl** the sequence is converged for trim $M = 2^{24}$ strings in 46 minutes while the equivalent tensor network code is not able to converge much past $n = 30$. In addition, if we consider $n = 30$ where both methods are converged, tensor networks is about 40 times slower than **PauliStrings.jl** for equivalent precision (considering BD=500 and trim $M = 2^{22}$). In addition, the convergence time and memory cost can be improved even more by taking advantage of the translation symmetry of the model, as explained in the Methods section. Doing this, we can now generate results for $N = 50$, converged up to $n = 50$, which are shown in Fig. 7 (A). Finally, we note that when unconverged the behavior is very different for both methods. The Pauli strings method tends to underestimate the correct sequence while tensor networks overestimates, and diverges. This allows for a much more controlled extrapolation of the correct result with increasing trim than for increasing bond dimension, and also reduces computation time when unconverged. Thus the Pauli strings method is much more efficient for this model, even without taking advantage of the translational symmetry.

We also consider the interacting **XXX** Heisenberg model which is integrable by the Bethe ansatz [48], and has the following Hamiltonian:

$$H = \sum_i \left( X_i X_{i+1} + Y_i Y_{i+1} + Z_i Z_{i+1} \right). \tag{21}$$

This Hamiltonian is constructed in **PauliStrings.jl** with the following code:

```
function XXX(N)
    H = ps.Operator(N)
    for j in 1:(N - 1)
```

```
        H += "X",j,"X",j+1
        H += "Y",j,"Y",j+1
        H += "Z",j,"Z",j+1
    end
    return H
end
```

We again consider open boundary conditions and use the initial operator $O_0 = \sum_j^N X_j Y_{j+1} - Y_j X_{j+1}$ which is shown to give a square root growth in Ref. [38]. The results are shown in Fig. 5 (B). In this case the methods are more comparable; the Pauli strings method is converged up to $n \sim 15$ while the tensor network method is converged up to $n \sim 17$, however, the latter requires more than an order of magnitude more time to reach this convergence than **PauliStrings.jl**. Taking advantage of the translation symmetry we achieve convergence up to $n \sim 20$, shown in Fig. 7 (B).

*Chaotic Model*

We now consider a chaotic chain which we call the **quantum Ising chain** with the following Hamiltonian:

$$H = \sum_i \left( X_i X_{i+1} - 1.05 Z_i + h_X X_i \right). \tag{22}$$

This code builds the Hamiltonian in **PauliStrings.jl**:

```
function Quantum_Ising(N, h_X)
    H = ps.Operator(N)
    for j in 1:(N - 1)
        H += "X",j,"X",j+1
    end
    for j in 1:N
        H += -1.05,"Z",j
        H += h_X,"X",j
    end
    return H
end
```

The Lanczos coefficients of generic chaotic systems grow linearly: $b_n \sim n$ (with a logarithmic correction in 1D $b_n \sim \frac{n}{\log n}$) [38, 49]. Here we use $h_X = 0.5$ which has been shown to be far away from the integrable point [50]. As an initial operator we use $O = \sum_i \left( 1.05 X_i X_{i+1} + Z_i \right)$, as is also used in Ref. [38].

The results are shown in Fig. 5 (C), where we see the expected growth $b_n \sim \frac{n}{\log n}$. Here the methods are again comparable, but tensor networks have an edge in convergence, converging up to $n = 40$ while **PauliStrings.jl** converges up to $n \sim 33$ in similar time. We thus conclude that the tensor network implementation is more efficient for this model, though the Pauli strings method is comparable. That being said, if we take advantage of the translational symmetry, shown in Fig. 7 (C) (and (D) for $h_X = 0.1$) the Pauli strings method now converges up to $n = 40$ in approximately half the time as the tensor network method.

A significant advantage of **PauliStrings.jl** is the relative ease of considering higher spatial dimensions. The Pauli string representation is not tied to any geometry; it allows us to work with local systems defined on arbitrary graphs. First we give an example of chaotic growth in 2D we using the following **2D XZ+ZX** model, with the following Hamiltonian:

$$H = \sum_{xy} \left( X_{x,y} Z_{x+1,y} + Z_{x,y} X_{x,y+1} \right) \tag{23}$$

We again use open boundary conditions and an initial operator $O = Z_{11}$. It has been proven that for this model the Lanczos coefficients grow linearly [38, 51], and we see this clearly in Fig. 6 (A). With **PauliStrings.jl** we are able to achieve up to $n = 15$ coefficients in a small amount of computation time (26 minutes). This competes with analytical methods used for computing Lanczos coefficients for other 2D models [37].

We also consider the **2D XXZ** model given by the following Hamiltonian:

$$H = \sum_{<i,j>} \left( X_i X_j + Y_i Y_j + \frac{1}{2} Z_i Z_j \right). \tag{24}$$

The Lanczos coefficients for this model are shown in Fig. 6. Convergence is achieved up to $n = 10$ in relatively small computation times.

## V.   VISUALIZING THE ALGEBRA

We now turn to a more pedagogical example in which the string representation gives direct intuitive insight into the system. Consider the XX model (eq. 20). This is an integrable model and if $O$ is a Majorana Pauli string, then $[H, O]$ is another Majorana string. A Majorana string is a string of the form $Y..YX1...1$ or $Y..YZ1...1$. Indeed, these strings anti-commute and can be interpreted as spin representations of Majorana fermions. Now, if we add a defect to the XX model, this breaks integrability. Let us visualize this effect using **PauliStrings.jl**. We already constructed the XX model below eq. (20). Now define a simple lanczos algorithm that prints the operator at each step :

```julia
function lanczos(H::ps.Operator, O::ps.Operator, steps::Int)
    O0 = deepcopy(O)
    b = ps.norm_lanczos(ps.com(H, O0))
    O1 = ps.com(H, O0)/b
    for n in 1:steps-1
        println("step ",n+1)
        println(O1)
        A = ps.com(H, O1)-b*O0
        b = ps.norm_lanczos(A)
        O = A/b
        O = ps.cutoff(O, 1e-10)
        O0 = deepcopy(O1)
        O1 = deepcopy(O)
```

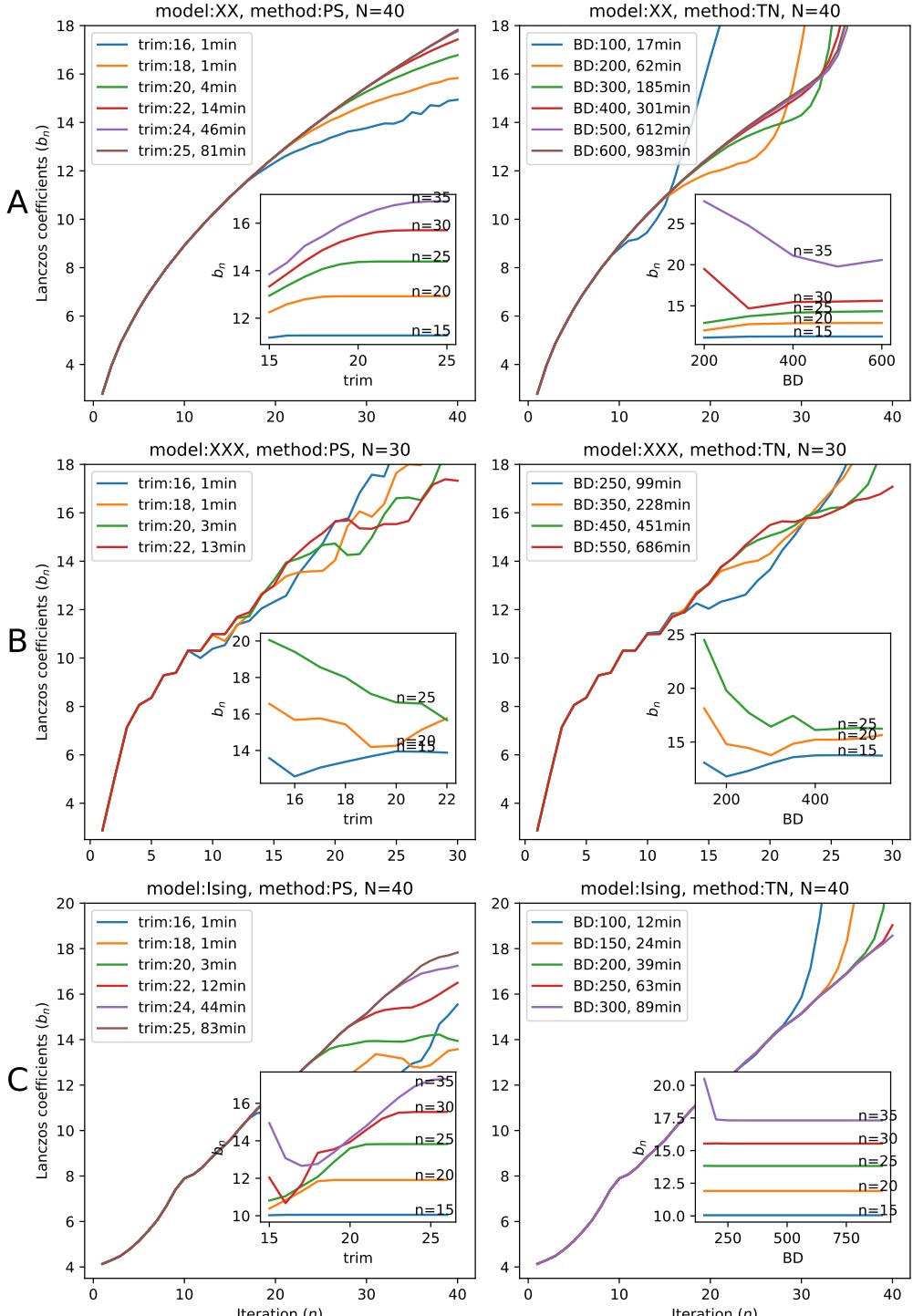

FIG. 5. Lanczos coefficients calculated with **PauliStrings.jl** (PS) and Tensor networks (TN) for the XX model (A), chaotic chain (B), and XXX model (C) for different trim and bond dimensions (BD) respectively. The trim value is $\log_2 M$ where $M$ is the maximum number of strings kept at each step. Similar results exploiting translation symmetry are shown in Fig. 7. Results are calculated with a maximum of 40 Gb memory and 1 CPU or 4 CPUs for **PauliStrings.jl** and tensor networks respectively. The insets show the convergence of the coefficients for some values of $n$.

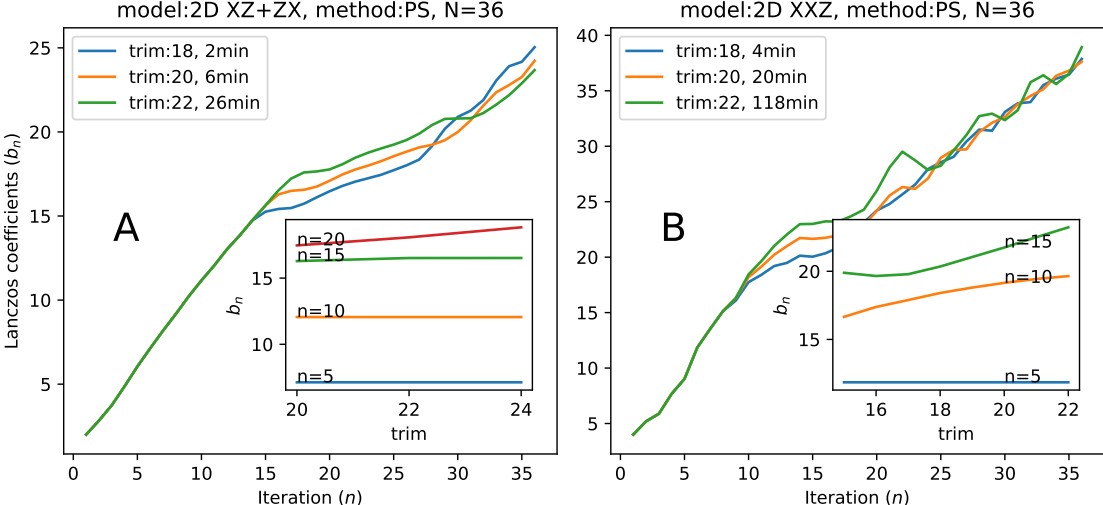

FIG. 6. Lanczos coefficients calculated with **PauliStrings.jl** for 2D chaotic models defined in equations (24) and (23).

```
        end
    end
```

Then we add a defect (a X field) on site 4 and run the Lanczos algorithm:

```
N = 10 #number of sites
H = XX(N) # construct a XX Hamiltonian
H += "X", 4 # add a defect on site 4
O = ps.Operator(N)
O += "X", 1
println(O)
lanczos(H, O, 7)
```

This yields the following output :

```
(1.0 + 0.0im) X111111111

step 2
(-0.0 + 1.0im) YZ11111111

step 3
(1.0 - 0.0im) YYX1111111

step 4
```

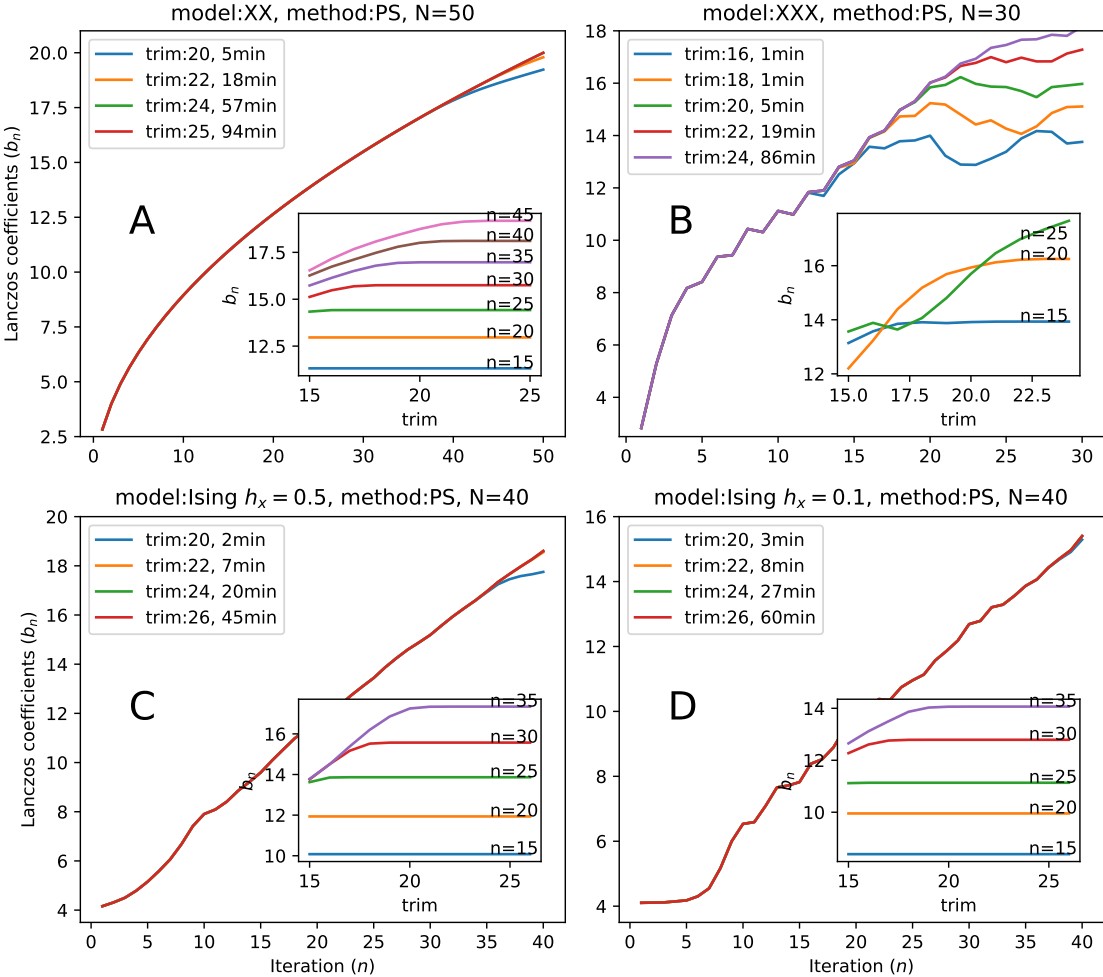

FIG. 7. Lanczos coefficients calculated with **PauliStrings.jl** exploiting translation symmetry. Instead of storing the whole operator at each step, we can take advantage of translation symmetry by only keeping strings that start on a particular site. This saves a factor $N$ in memory and allows better convergence for similar memory usage.

```
(-0.0 + 1.0im) YYYZ111111

step 5
(0.7071067812 + 0.0im) YYYY111111
(0.7071067812 + 0.0im) YYYYX11111

step 6
(-0.0 + 0.8164965809im) YYYZX11111
(-0.0 - 0.4082482905im) YYYXZ11111
(0.0 + 0.4082482905im) YYYYYZ1111
```

```
step 7
(0.4 - 0.0im) YYX1X11111
(-0.6 + 0.0im) YYY1Y11111
(-0.2 + 0.0im) YYYXYX1111
(0.2 - 0.0im) YYZ1Z11111
(-0.6 + 0.0im) YYYZYZ1111
(0.2 - 0.0im) YYYYYYX111
```

In the beginning, each Majorana is transformed into a single other Majorana, and the string grows until it hits the defect at step 5. Then the defect breaks integrability and the number of strings starts exploding. That's a simple example of how the Pauli string method can also be used as a pedagogical and insightful way to visualize physical phenomena such as integrability-breaking.

## VI.    CONCLUSION

We have shown that **PauliStrings.jl** provides a competitive platform for studying quantum many-body dynamics. Examples of this are presented for Heisenberg time evolution and Krylov subspace expansion through the recursion method. One of the important strengths of Pauli strings is that they provide a natural framework to take advantage of noise to make simulations tractable. In addition, though tensor network methods quickly break down with increasing long-range entanglement, some systems with this type of entanglement can still be decomposed into a small number of strings, making Pauli strings more efficient for these kinds of systems. Furthermore, Pauli string methods are not as limited in spatial dimension and geometry, and arbitrary geometries are easy to implement. **PauliStrings.jl** is easily installable through the Julia language package manager, and more thorough code examples can be found in the documentation [52].

However, right now the truncation schemes that we use are very basic. In the Lanczos case, we just discard strings with the smallest weight. In the time evolution case, we first add noise and then discard strings with smallest weight, effectively discarding long strings that are more affected by noise. There may be more efficient truncation schemes, and indeed, in certain cases, long strings do matter. For example, in the $XX$ model, even if the model is integrable, nested commutators generate a few very long strings, as shown in the previous section. This suggests that in this model, discarding long strings is not the best strategy. Ideally we would like to predict what strings matter and what strings don't. A more refined heuristic truncation scheme would be able to estimate the impact of discarding a string on the higher-order nested commutators. One idea would be to use machine learning techniques to predict the importance of strings.

Also note that our current implementation is not yet parallelized yet beats parallelized tensor networks codes in many cases. Parallelization would offer a straightforward route to improvement. In the future, one valuable application of the Pauli String method would be a technique that lets us probe spectral properties. For example, an interesting quantity to compute is the thermal average $\frac{\mathrm{Tr}(e^{-\beta H}O)}{\mathrm{Tr}(e^{-\beta H})}$. In the large $\beta$ limit and when $O$ is the Hamiltonian itself, this converges to the ground-state energy. An approach to computing such a quantity is to expand it into connected moments [53–56]. Strings are particularly appropriate for computing moments since they take advantage of the sparsity of the operators. Moreover, computing the moment $\mathrm{Tr}\, H^k$ does not require storing $H^k$. For example if $H = \sum_i h_i \tau_i$ then the 4th moment is $\mu_4 = \sum_{ijkl} h_i h_j h_k h_l \, \mathrm{Tr}(\tau_i \tau_j \tau_k \tau_l)$ and computing $\mu_4$ only requires accumulating terms $h_i h_j h_k h_l$ such that $\tau_i \tau_j \tau_k \tau_l = \mathbb{1}$. From a more foundational point of view, we also have suggested that the moments of local Hamiltonians may give us insight

on the decomposition into subsystems and the quantum to classical transition [57]. An alternative to moment expansion for estimating ground state expectation values is to imaginary-time-evolve $H$ and $O$.

## ACKNOWLEDGMENTS

N.L. was supported by a research grant (42085) from Villum Fonden. N.L. thanks Prof. Berislav Buca for funding and support, which made finishing this project possible. D.S. and J.C.P. are partially supported by AFOSR (Award no. FA9550-25-1-0067) and NSF (Award no. OAC-2118310). This work was supported in part through the NYU IT High Performance Computing resources, services, and staff expertise.

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
