# Peer review of "Quantum many-body simulations with PauliStrings.jl"

_SciPost Physics Codebases, doi:SciPost Phys. Codebases 54-r1.5 (2025) , SciPost Phys. Codebases 54 (2025)_

## Round 1 · Referee Report · Anonymous (Referee 1) · 2025-1-14

Strengths

1- PauliStrings.jl provides an efficient implementation of operator time evolution based on binary representations of the Pauli group. 2- Efficiency is demonstrated by benchmarking against MPS simulations for the particularly challenging case of dynamics at infinite temperature and against MPO-based simulations for the computation of Lanczos coefficients. 3- The library provides a compact and intuitive interface to define the system and run simulations.

Weaknesses

1- Computing a time-dependent two-point correlation function of a spin chain at infinite temperature is considered as a benchmark case. It is demonstrated, that the results qualitatively agree with the expectation of diffusive dynamics. A more quantitative assertion of the quality of the results could have been possible by extracting a diffusion constant for a model, where reference values exist.

2- Parts of the manuscript are hard to read, because they mostly consist of figures and code listings (especially section 5). It would be good to organize the arrangement such that the figures appear as close as possible to their reference in the main text and avoid their interleaving with code listings.

Report

The manuscript describes a Julia library for quantum operator time evolution based on Pauli strings. This approach received quite some interest recently and it has been shown to yield competitive results when compared to other state-of-the-art methods in a number of use cases. The availability of a highly efficient implementation in the form of a Julia library is therefore very valuable for the community.

The manuscript reports two relevant example applications and runtime benchmarks in comparison with tensor network simulations. The source code is clean, well documented, and tested. The library is accompanied by clear documentation. Therefore, the acceptance criteria are met and I recommend publication, once the comments and requested changes below have been considered for a minor revision.

Comments:

  • First sentence in second paragraph of "Tensor networks": The sentence mentions MPS and MPO, but only describes the shape of MPS tensors, which is not clear. It would be good to clarify, that the three-leg-tensor is the MPS building block, not MPO.
  • In the manuscript, the inner product tr(A^\dagger B)/2^N is called "infinite temperature inner product". I think Frobenius inner product (up to normalization) would be a more appropriate term.

Requested changes

1- In Algorithm 1, there are two subsequent assignments to a variable 'v'. Should one of them be a 'w'? Moreover, the notation if (u,v) is in C, then C[(u,v)] <- C[(u,v)] + h is not particularly clear. For example, above that, the operators A and B seem to store h-values in A.h or B.h member variables. But here the value is directly obtained from and assigned to C. It would be good to clarify this. 2- In Algorithm 2, the notation is chosen different than algorithm 1. For example, in Algorithm 1, double indices appear explicitly as tuples or independent member variables of the operators, whereas they are a single variable in Algorithm 2. Moreover, I assume, that the dot appearing in the argument of 'Shiftleft' is the product function from Algorithm 1, but that's not clear. It would be good to make the notation in the different Algorithms clear and consistent.

Recommendation

Ask for minor revision

  • validity: high
  • significance: high
  • originality: high
  • clarity: good
  • formatting: acceptable
  • grammar: excellent

Author:  Nicolas Loizeau  on 2025-01-30  [id 5162]

(in reply to Report 1 on 2025-01-14)

Dear Referee,
Thank you for your time and feedback. We have addressed all the requested changes. In addition, we agree that extracting the diffusion constants for a few models using the Pauli strings method would be a valuable benchmark and could by itself be an exciting project. The main goal of this paper is to present the PauliString package through a few simple and accessible examples so we decided to not focus too much on hydrodynamics, but we will likely explore this in the future.

---

## Round 1 · Referee Report · Johannes Hauschild (Referee 2) · 2025-1-21

Strengths

  1. Efficient representation of operators that are sums of pauli strings
  2. Clear presentation of benchmarks
  3. Clear documentation of the package with several examples.

Weaknesses

Notation in section II is a bit sloppy and partially unclear.

Report

The manuscript presents the software library PauliStrings.jl which implements an efficient binary representation of opeartors which are sums of Pauli strings. They show clear benchmarks and demonstrations of its capabilities, comparing to MPS/MPO time evolution using ITensor both in terms of values and runtime.
With Fig. 4, they also clearly show that the sparsity of the Pauli string representation can help to reach significantly longer times with good precision especially in 2D, making the library a useful tool for exploring operator/hydro-dynamics at least at infinite temperatures. (Whether the sparsity is still preserved at finite temperatures, as suggested as a possible application in the conclusions, remains to be seen.)

Requested changes

  1. The notation and equations in section II should be double-checked:
  2. In Equation 7 and 11, you have factors $(-1)^{\alpha_{1,2}}$ which you don't introduce and take into account for fthe phase of eq. 10. Further, to be consistent with $10$, the phase on the right hand side should be just $\alpha_{12}$, not $(-1)^{\alpha_{12}}$
  3. given that you write the explicit formula for $\alpha_{12}$ in (10), you could also explicitly write $\alpha = (-1)^{i \wedge l }$ just before equation (6).
  4. Equation 5 is missing a minus sign to be consitent with (7-10) (assuming that drop the phases on the left side of (7))
  5. Why not use $v_1, w_1, v_2, w_2$ for the bits in equation 6 as well? As mentioned by the other referee already, there's also a typo in algorithm 1, with a v instead of w.

  6. In algorithm 2, it should be made clear what $N$ is - if I'm not mistaken, it should be $min(\mathrm{StringLength}(A.\tau_i), \mathrm{StringLength}(B.\tau_i))$, where $\mathrm{StringLength}$. Maybe also add a small comment that it is very straightforward to generalize Algorithm 2 to higher dimensions as well by iterating over the necessary shifts (x,y) instead of just x.

Recommendation

Publish (easily meets expectations and criteria for this Journal; among top 50%)

  • validity: high
  • significance: high
  • originality: high
  • clarity: top
  • formatting: perfect
  • grammar: perfect

Author:  Nicolas Loizeau  on 2025-01-30  [id 5163]

(in reply to Report 2 by Johannes Hauschild on 2025-01-21)

Dear Dr Hauschild
Thank you for your time and feedback and in particular for pointing out all of the issues with section 2. There were indeed multiple typos in the equations that we have now corrected.

---

## Round 2 · Referee Report · Anonymous (Referee 1) · 2025-2-18

Report

All my previous remarks have been addressed in a satisfactory manner.

I would only suggest to speak of rank-3 (or rank-4) instead of 3-dimensional (or 4-dimensional) tensors in the "Tensor networks" section.

Recommendation

Publish (easily meets expectations and criteria for this Journal; among top 50%)

---

## Round 2 · List of Changes

• Equations (6,7,10,11) in section 2 have been corrected
  • Algorithms 1 and 2 have been rewritten more clearly
  • The second paragraph of 'Tensor networks' has been updated to clarify the notions of MPS and MPO
  • A few references have been added
  • infinite temperature inner product - > Frobenius inner product
  • Add the sentence : "Note that Algorithm 2 can easily be adapted to higher dimensions by iterating over the necessary shifts corresponding to each dimension."

---

## Editorial Decision

published